# Learning to Noise: Application-Agnostic Data Sharing with Local Differential Privacy

## Abstract

In recent years, the collection and sharing of individuals' private data has become commonplace in many industries. *Local* differential privacy (LDP) is a rigorous approach which uses a randomized algorithm to preserve privacy even from the database administrator, unlike the more standard *central* differential privacy. For LDP, when applying noise directly to high-dimensional data, the level of noise required all but entirely destroys data utility. In this paper we introduce a novel, application-agnostic privatization mechanism that leverages representation learning to overcome the prohibitive noise requirements of direct methods, while maintaining the strict guarantees of LDP. We further demonstrate that data privatized with this mechanism can be used to train machine learning algorithms. Applications of this model include private data collection, private novel-class classification, and the augmentation of clean datasets with additional privatized features. We achieve significant gains in performance on downstream classification tasks relative to benchmarks that noise the data directly, which are state-of-the-art in the context of application-agnostic LDP mechanisms for high-dimensional data sharing tasks.

## 1 Introduction

The collection of personal data is ubiquitous, and unavoidable for many in everyday life. While this has undeniably improved the quality and user experience of many products and services, evidence of data misuse and data breaches (Sweeney, 1997; Jolly, 2020) have brought the concept of data privacy into sharp focus, fueling both regulatory changes as well as a shift in personal preferences. The onus has now fallen on organizations to determine if they are willing and able to collect personal data under these changing expectations. There is thus a growing need to collect data in a privacy-preserving manner, that can still be used to improve products and services.

Often coined the 'gold standard' of privacy guarantees, central differential privacy (CDP) (Dwork & Roth, 2014) protects against an adversary determining the presence of a user in a dataset. It provides a quantifiable definition, is robust to post-processing, and allows for the protection of groups of people via its composability property (Dwork et al., 2006; Dwork & Roth, 2014). The CDP framework relies on the addition of noise to the output of statistical queries on a dataset, in such a way that the same information can be extracted from that dataset whether or not any given individual is present.

One can train machine learning models such that the model is CDP with respect to the training set by using training methods such as DP-SGD (Abadi et al., 2016), DP-Adam (Gylberth et al., 2017) or PATE (Papernot et al., 2017). Given access to a clean labelled dataset, one could train, for example, a CDP classifier this way. Similarly, one could 'share' data privately by training a generative model, such as a variational autoencoder (VAE) or generative adversarial network (GAN), with a CDP training method, and then construct a synthetic dataset satisfying CDP with respect to the training set by generating samples from this model (Xie et al., 2018; Triastcyn & Faltings, 2019; Acs et al., 2019; Takagi et al., 2021). While some of these approaches require only an unlabelled training set, their applications are limited in that they generate only synthetic samples that are likely under the original training set distribution. Firstly, changes in the data distribution warrant the re-training of the generative model. Secondly, the synthetic points are only representative samples, and so we lack any information about the features of given individuals. Clearly, this limits the range of applications: the model cannot be used to join private and clean datasets, for applications in which

we expect a distributional shift, or for privately collecting new data. Furthermore, CDP also requires a trustworthy database administrator.

Federated learning (McMahan et al., 2017; Agarwal et al., 2018; Rodríguez-Barroso et al., 2020) is a related technique in which a model can be trained privately, without placing trust in a database administrator. The model is trained across multiple decentralized devices, omitting the need to share data with a central server. Unfortunately, by construction, federated learning does not allow for any data to be privately shared or collected, and is only concerned with the privacy of the model.

An approach that allows data collection, while protecting the privacy of an individual against even the database administrator, is to construct a mechanism that privatizes the features of a given individual locally, before collection. Warner (1965) developed a mechanism, known as randomized response, to preserve the privacy of survey respondents: when answering a sensitive (binary) question, the respondent is granted plausible deniability by giving a truthful answer if a fair coin flip returns heads, and answering yes otherwise.

Recent work has further developed this idea, often referred to as local differential privacy (LDP) (Kasiviswanathan et al., 2008; Duchi et al., 2013). LDP provides a mathematically-provable privacy guarantee for members of a database against both adversaries and database administrators. Many existing LDP mechanism focus on the collection of low-dimensional data like summary statistics, or on mechanisms which do not easily generalize to different data types. Erlingsson et al. (2014), Ding et al. (2017) and Tang et al. (2017) introduce methods for collecting such data repeatedly over time. Erlingsson et al. (2014) find that for one time collection, directly noising data (after hashing to a bloom filter) is their best approach for inducing privacy. Ren et al. (2018) extend this bloom filter based approach, and attempt to estimate the clean distribution of their LDP data in order to generate a LDP synthetic dataset, though we note that the range of applications is limited, as with CDP synthetic generation above.

In this paper, we adapt well-established techniques from representation learning to address the fundamental limitation of LDP in the context of high-dimensional data: datapoints in high dimensional space require prohibitive levels of noise to locally privatize (the privacy budget, $\epsilon$, naively scales linearly with the dimensionality). To motivate our approach, consider the fact that it is often a good approximation to assume that a given high-dimensional dataset lives on a much lower dimensional manifold. Applying a general privatization mechanism to low-dimensional representations should thus enable us to *learn* how to add noise to the high-dimensional data efficiently and application-agnostically. Our approach is inspired by the VAE (Kingma & Welling, 2014; Rezende et al., 2014); we demonstrate that sampling in latent space is equivalent to passing a datapoint through an LDP Laplace mechanism. Furthermore, reconstructing a datapoint is equivalent to adding a complex noise distribution to the raw features, thereby inducing LDP.

Our randomized algorithm, which we refer to as the variational Laplace mechanism (VLM), satisfies the strict guarantees of LDP, and is agnostic to both data type and downstream task. We demonstrate that we can use the data privatized with our mechanism to train downstream machine learning models that act on both clean and privatized data at inference time. Furthermore, we demonstrate multiple concrete applications of our model: we privately collect data from individuals for downstream model training; we use a transfer-learning-inspired approach to privately collect data of an unseen class type upon which we train a classifier; and we augment a clean dataset with additional privatized features to improve the accuracy of a classifier on the combined data. None of these applications can be solved with CDP, and we find significant performance gains over the naive approach of directly noising the data.

## 2 BASIC DEFINITIONS AND NOTATION

To formalize the concept of differential privacy, we first introduce some definitions and notation.

**Definition (($\epsilon, \delta$)-central differential privacy):** Let $\mathcal{A} : \mathcal{D} \to \mathcal{Z}$ be a randomized algorithm, that takes as input datasets from the dataset domain $\mathcal{D}$. We say $\mathcal{A}$ is ($\epsilon, \delta$)-central differentially private if for $\epsilon, \delta \geq 0$, for all subsets $S \subseteq \mathcal{Z}$, and for all neighboring datasets $D, D' \in \mathcal{D}$, we have

$$p(\mathcal{A}(D) \in S) \ \leq \ \exp(\epsilon)\, p(\mathcal{A}(D') \in S) + \delta \tag{1}$$

where for $D$ and $D'$ to be neighboring means that they are identical in all but one datapoint.

Intuitively, this states that one cannot tell (with a level of certainty determined by $(\epsilon, \delta)$) whether an individual is present in a database or not. When $\delta = 0$ we say $\mathcal{A}$ satisfies $\epsilon$-CDP.

**Definition ($\ell_1$ sensitivity):** The $\ell_1$ sensitivity of a function $f : \mathcal{D} \to \mathbb{R}^k$ is defined as

$$\Delta f = \max_{\text{adjacent}(D, D')} ||f(D) - f(D')||_1 \qquad (2)$$

where adjacent$(D, D')$ implies $D, D' \in \mathcal{D}$ are neighboring datasets.

**Definition (Laplace mechanism):** The Laplace mechanism $\mathcal{M}^{(\text{central})} : \mathcal{D} \to \mathbb{R}^k$ is a randomized algorithm defined as

$$\mathcal{M}^{(\text{central})}(D, f(\cdot), \epsilon) = f(D) + (s_1, \dots, s_k) \qquad (3)$$

for $D \in \mathcal{D}$, $s_i \sim \text{Laplace}(0, \Delta f/\epsilon)$, and some transformation function $f : \mathcal{D} \to \mathbb{R}^k$.

The Laplace mechanism induces $\epsilon$-CDP; see Dwork & Roth (2014) for proof.

While CDP relies on a trusted database administrator, LDP provides a much stricter guarantee in which the individual does not need to trust an administrator. Instead individuals are able to privatize their data before sending it using a *local* randomized algorithm.

**Definition (($\epsilon, \delta$)-local differential privacy):** A local randomized algorithm $\mathcal{A} : \mathcal{X} \to \mathcal{Z}$, that takes as input a datapoint from the data domain $\mathcal{X}$, satisfies $(\epsilon, \delta)$-local differential privacy if for $\epsilon, \delta \geq 0$, for all $S \subseteq \mathcal{Z}$, and for any inputs $x, x' \in \mathcal{X}$,

$$p(\mathcal{A}(x) \in S) \leq \exp(\epsilon)\, p(\mathcal{A}(x') \in S) + \delta \qquad (4)$$

When $\delta = 0$ we say $\mathcal{A}$ satisfies $\epsilon$-LDP.

**Definition (Local Laplace mechanism):** The local Laplace mechanism $\mathcal{M}^{(\text{local})} : \mathcal{X} \to \mathbb{R}^k$ is a randomized algorithm defined as

$$\mathcal{M}^{(\text{local})}(x, f(\cdot), \epsilon) = f(x) + (s_1, \dots, s_k) \qquad (5)$$

for $x \in \mathcal{X}$, $s_i \sim \text{Laplace}(0, \Delta f/\epsilon)$, and some transformation function $f : \mathcal{X} \to \mathcal{Z}$, where $\mathcal{Z} \subseteq \mathbb{R}^k$ and the $\ell_1$ sensitivity of $f(\cdot)$ is defined as $\Delta f = \max_{x, x' \in \mathcal{X}} ||f(x) - f(x')||_1$.

The local Laplace mechanism satisfies $\epsilon$-LDP (see Appendix A for proof).

Another common choice of mechanism for privatizing continuous data is the Gaussian mechanism, which satisfies $(\epsilon, \delta > 0)$-LDP. For the remainder of the paper however, we exclusively study the local Laplace mechanism since it provides a strong privacy guarantee (i.e. $\delta = 0$). We note that our approach could be used to learn a Gaussian mechanism with minimal modification.

## 3 PROPOSED METHOD

Early work on data collection algorithms, such as randomized response, have relied on very simple randomized algorithms to induce privacy. Throughout this paper we have benchmarked our results against such a mechanism, in which we add Laplace noise to all continuous features, and flip each of the categorical features with some probability. By the composition theorem (Dwork & Roth, 2014), each feature then contributes towards the overall LDP guarantee of the $d$-dimensional datapoint as $\epsilon = \sum_{i=1}^{d} \epsilon_i$. Since we have no prior knowledge about which features are most important, we choose the Laplace noise level (or flip probability) for each feature to be such that $\epsilon_i = \epsilon/d$. See Appendix E.2 for further details. As with our approach, this benchmark mechanism can act on any date type, and forms a downstream task agnostic LDP version of the data.

As $d$ increases, $\epsilon_i$ decreases for each feature $i$. The noise required to induce $\epsilon$-LDP thus grows with data dimensionality. For high-dimensional datasets like images or large tables, features are often highly correlated; consequently, noising features independently is wasteful towards privatizing the information content in each datapoint. A more effective approach to privatization involves noising a learned lower-dimensional representation of each datapoint using a generic noising mechanism.

To this end, we use a VAE-based approach to learn a low-dimensional latent representation of our data. This learned mapping from data space to latent space forms our function $f(\cdot)$ in Equation 5, and requires only a small unlabelled dataset from a similar distribution, which most organizations

will typically already have access to, either internally or publicly. Applying the Laplace mechanism (as described in Section 3.1) thus ensures the encoded latents, as well as reconstructed datapoints, satisfy LDP. We can therefore privatize data at the latent level or the original-feature level; preference between these two options is application specific, and we investigate both experimentally.

Data owners can apply this learned LDP mechanism to their data before sharing, and the database administrator forms an LDP data set from the collected data. This set, along with information on the type of noise added, is used to train downstream machine learning algorithms. Though our privatization method is task agnostic, in this paper we focus on classification tasks in which we have some features $x$ for which we want to predict the corresponding label $y$. At inference time, we show that this classifier can act on either clean or privatized datapoints, depending on the application.

## 3.1 LEARNING A LAPLACE MECHANISM

We assume that our data $x$ is generated by a random process involving a latent variable $z$. We then optimize a lower bound on the log likelihood (Kingma & Welling, 2014)

$$\log p(x) = \log \int p(z) p_\theta(x|z) dz \geq \mathbb{E}_{q_\phi(z|x)} \left[ \log p_\theta(x|z) \right] - D_{\mathrm{KL}} \big( q_\phi(z|x) || p(z) \big) \qquad (6)$$

where $p(z)$ is the prior distribution and $q_\phi(z|x)$ is the approximate posterior. The generative distribution $p_\theta(x|z)$ and approximate inference distribution $q_\phi(z|x)$ are parameterized by neural networks, with learnable parameters $\theta$ and $\phi$ respectively. While the distributions over latent space are commonly modeled as Gaussian, we aim to learn a Laplace mechanism and so instead we choose

$$p(z) = \prod_{i=1}^{d} p(z_i), \quad \text{and} \quad q_\phi(z|x) = \prod_{i=1}^{d} q_\phi(z_i|x) \qquad (7)$$

where $p(z_i) \sim \mathrm{Laplace}(0, 1/\sqrt{2})$ and $q_\phi(z_i|x) \sim \mathrm{Laplace}(\mu_\phi(x)_i, b)$.

We parameterize $\mu_\phi(\cdot)$ with a neural network and restrict its output via a carefully chosen activation function $\nu(\cdot)$ acting on the final layer $\mu_\phi(.) = \nu(h_\phi(\cdot))$. This clips the output $h_\phi(\cdot)$ such that all points are within a constant $\ell_1$-norm $l$ of the origin, by re-scaling the position vector of points at a larger $\ell_1$ distance. In this way we ensure that $\Delta\mu_\phi = 2l$.

With $\Delta\mu_\phi$ finite, we note that if we fix the scale $b = 2l/\epsilon_x$ at inference time, then drawing a sample from our encoder distribution $q_\phi(z|x)$ is equivalent to passing a point $x$ through the Local Laplace mechanism $\mathcal{M}^{(\mathrm{local})}(x, \mu_\phi(\cdot), \epsilon_x)$ from Equation 5. Therefore to obtain a representation $\tilde{z}$ of $x$ that satisfies $\epsilon_x$-LDP, we simply have to pass it through the encoder mean function $\mu_\phi(\cdot)$ and add $\mathrm{Laplace}(0, 2l/\epsilon_x)$ noise. We refer to this model as a variational Laplace mechanism (VLM).

We further prove in Appendix B that a reconstruction $\tilde{x}$ obtained by passing $\tilde{z}$ through the decoder network also satisfies $\epsilon_x$-LDP, allowing us to privatize datapoints at either latent level $\tilde{z}$, or original-feature level $\tilde{x}$.

Note that $b$ is always fixed at inference (i.e. data privatization) time to guarantee $\tilde{z}$ is $\epsilon_x$-LDP. However, we experiment with learning $b$ during training, as well as fixing it to different values.

Certain applications of our model require us to share either the encoder or decoder of the VLM at inference time. If the VLM training data itself contains sensitive information, then the part of the network that gets shared must satisfy CDP with respect to this training data. We found the following two-stage VLM training approach to be helpful in these cases:

- **Stage 1:** Train a VLM with encoding distribution $q_\phi(z|x)$ and decoding distribution $p_\theta(x|z)$ using a non-DP optimizer, such as Adam (Kingma & Ba, 2015).

- **Stage 2:** If training a DP-encoder model, fix $\theta$, and re-train the encoder with a new distribution $q_{\phi_{\mathrm{private}}}(z|x)$. If training a DP-decoder, fix $\phi$ and replace the decoder with $p_{\theta_{\mathrm{private}}}(x|z)$. Optimize $\phi_{\mathrm{private}}$ or $\theta_{\mathrm{private}}$ as appropriate using DP-Adam (Gylberth et al., 2017).

Section 4 and Appendix D outline applications in which private VLM components are required.

### 3.2 Training on Private Data

For supervised learning we must also privatize our target $y$. For classification, $y \in \{1, \ldots, K\}$ is a discrete scalar. To obtain a private label $\tilde{y}$, we flip $y$ with some fixed probability $p < (K-1)/K$ to one of the other $K-1$ categories: $p(\tilde{y} = i | y = j) = (1-p)\mathbb{I}(i=j) + p/(K-1)\mathbb{I}(i \neq j)$. Setting $p = (K-1)/(e^{\epsilon_y} + K - 1)$ induces $\epsilon_y$-LDP (see Appendix C for proof).

By the composition theorem (Dwork & Roth, 2014), the tuple $(\tilde{x}, \tilde{y})$ satisfies $\epsilon$-LDP where $\epsilon = \epsilon_x + \epsilon_y$. Downstream models may be more robust to label noise than feature noise, or vice versa, so for a fixed $\epsilon$ we set $\epsilon_x = \lambda\epsilon$ and $\epsilon_y = (1 - \lambda)\epsilon$, where $\lambda$ is chosen to maximise the utility of the dataset. In practice, we treat $\lambda$ as a model hyperparameter.

Rather than training the classifier $p_\psi(\cdot)$ directly on private labels, we incorporate the known noise mechanism into our objective function

$$\log p(\tilde{y}|\tilde{x}) = \log \sum_{y=1}^{K} p(\tilde{y}|y)\, p_\psi(y|\tilde{x}) \quad \text{or} \quad \log p(\tilde{y}|\tilde{z}) = \log \sum_{y=1}^{K} p(\tilde{y}|y)\, p_\psi(y|\tilde{z}) \tag{8}$$

depending on whether we choose to work on a feature level, or latent level.

At inference time we can classify privatized points using $p_\psi(y|\tilde{x})$ or $p_\psi(y|\tilde{z})$. We also show empirically that we can classify clean points using the same classifier. We refer to these tasks as *private* and *clean* classification, with applications given in Sections 4 and 5.

### 3.3 Private Validation and Hyper-parameter Optimization

Typically for model validation, one needs access to clean labels $y$ (and clean data $x$ for validating a clean classifier). However, we note that we need only collect privatized model performance metrics on test and validation sets, rather than actually collect the raw datapoints.

To do this, we send the trained classifier to members of the validation set so that they can test whether it classified correctly: $c \in \{0, 1\}$. They return an answer, flipped with probability $p = 1/(e^\epsilon + 1)$ such that the flipped answer $\tilde{c}$ satisfies $\epsilon$-LDP, and we estimate true validation set accuracy $A = \frac{1}{N_{\text{val}}}\sum_{n=1}^{N_{\text{val}}} c_n$ from privatized accuracy $\tilde{A} = \frac{1}{N_{\text{val}}}\sum_{n=1}^{N_{\text{val}}} \tilde{c}_n$ using

$$A = \frac{\tilde{A} - p}{1 - 2p} \tag{9}$$

(Warner, 1965). We use this method to implement a grid search over hyperparameters of our model.

## 4 Classifying Clean Datapoints: Applications and Experiments

Below we demonstrate the versatility of our model by outlining a non-exhaustive list of potential applications, with corresponding experiments. Experiments are trained on the MNIST dataset (Le-Cun et al., 1998), or the Lending Club dataset[1]. The CDP requirements differ between applications, but are explicitly stated for each application in Appendix D. For all stated $(\epsilon, \delta)$-CDP results we use $\delta = 10^{-5}$, whilst for $(\epsilon, \delta)$-LDP results, $\delta = 0$. All results quoted are the mean of 3 trials; error bars represent $\pm 1$ standard deviation. Appendix E describes experimental setup and dataset information.

In Sections 4.1 and 4.2, we investigate the *clean* classification task, and report on *clean* accuracy. Namely, the classifiers are trained on privatized data in order to classify clean datapoints at inference time. We also study the classification of privatized datapoints, using a classifier trained on privatized data, which we refer to as *private* classification (and report *private* accuracy) in Section 5.

### 4.1 Data Collection

Organizations may have access to some (potentially unlabelled) clean, internal data $\mathcal{D}_1$, but want to collect privatized labelled data $\mathcal{D}_2$ in order to train a machine learning algorithm. For example, a public health body may have access to public medical images, but want to train a diagnosis classifier

---

[1] https://www.kaggle.com/wordsforthewise/lending-club

to be used in hospitals using labelled data collected privately from their patients. Similarly, a tech company with access to data from a small group of users may want to train an in-app classifier; to do so they could collect private labelled training data from a broader group of users, before pushing the trained classifier to the app. Finally, a multinational company may be allowed to collect raw data on their US users, but only LDP data on users from countries with more restrictive data privacy laws.

In this experiment, we split the data such that $\mathcal{D}_1$ and $\mathcal{D}_2$ follow the same data distribution, however in practice this may not always be the case. For example, when $\mathcal{D}_2$ is sales data collected in a different time period, or user data collected in a different region, we may expect the distribution to change. We have omitted such an experiment here, but the extreme case of this distributional shift is explored experimentally in Section 4.2.

We run this experiment on both MNIST and Lending Club. As in Sections 3.1 - 3.3, we first train a VLM with a DP encoder using $\mathcal{D}_1$, then privatize all datapoints and corresponding labels in $\mathcal{D}_2$ before training a classifier on this privatized training set. Results are shown in Figure 1.

For both MNIST and Lending Club, we significantly outperform the baseline approach of noising pixels directly. The benchmark performed at random accuracy for all local $\epsilon \leq 100$ for MNIST (local $\epsilon \leq 20$ for Lending Club). Our model performed well above random for all local $\epsilon$ values tested.

For MNIST, we see that latent-level classification outperforms feature-level classification for higher local-$\epsilon$ values. Indeed, the data processing inequality states we cannot gain more information about a given datapoint by passing it through our decoder. However at lower local $\epsilon$, we see feature level classification accuracy is higher. We hypothesize that at this point, so much noise is added to the latent that the latent level classifier struggles, while on pixel level the VLM decoder improves classification by adding global dataset information from $\mathcal{D}_1$ to the privatized point.

For Lending Club, we do not see a clear difference between latent-level and feature-level accuracy. However we also note that the features are not as highly correlated as in MNIST, so perhaps the decoder has less influence on results.

Finally, we see that reducing central $\epsilon$ has an adverse effect on MNIST classification accuracy, especially for higher local $\epsilon$. The effect seems negligible for Lending Club and we hypothesize that this is due to the large quantity of training data, along with the easier task of binary classification.

## 4.2 NOVEL-CLASS CLASSIFICATION

As discussed in Section 4.1, the internal data $\mathcal{D}_1$ and the data to be collected $\mathcal{D}_2$, may follow different data distributions. In the extreme case, the desired task on $\mathcal{D}_2$ may be to predict membership in a class that is not even present in dataset $\mathcal{D}_1$. For example, in a medical application there may be a large existing dataset $\mathcal{D}_1$ of chest scans, but only a relatively small dataset $\mathcal{D}_2$ that contains patients with a novel disease. As before, a public health body may want to train a novel-disease classifier to distribute to hospitals. Similarly, a software developer may have access to an existing dataset $\mathcal{D}_1$, but want to predict software usage data for $\mathcal{D}_2$, whose label is specific to the UI of a new release.

We run this experiment on MNIST, where the internal $\mathcal{D}_1$ contains training images from classes 0 to 8, (with a small number of images held out for classification), and $\mathcal{D}_2$ contains all training images from class 9. As in Sections 3.1 and 3.2, we first train a VLM with a DP encoder on $\mathcal{D}_1$, then privatize all images in $\mathcal{D}_2$ (we are not required to collect or privatize the label since all images have the same label). We then train a binary classifier on the dataset formed of the private 9's and the held out internal images from classes 0-8 (which we privatize and label 'not 9's').

Results are shown in Figure 2. On both latent and feature level, we obtain approximately 75% accuracy at local $\epsilon = 2$ and above random performance at local $\epsilon = 1$. The benchmark of noising features directly achieves random performance for all local $\epsilon \leq 100$ (not shown in the figure). Once again, the effects of reducing central $\epsilon$ appear greater at higher local $\epsilon$ values, and we see that the latent-level classifier outperforms the feature-level classifier at high local $\epsilon$, but not at lower local $\epsilon$.

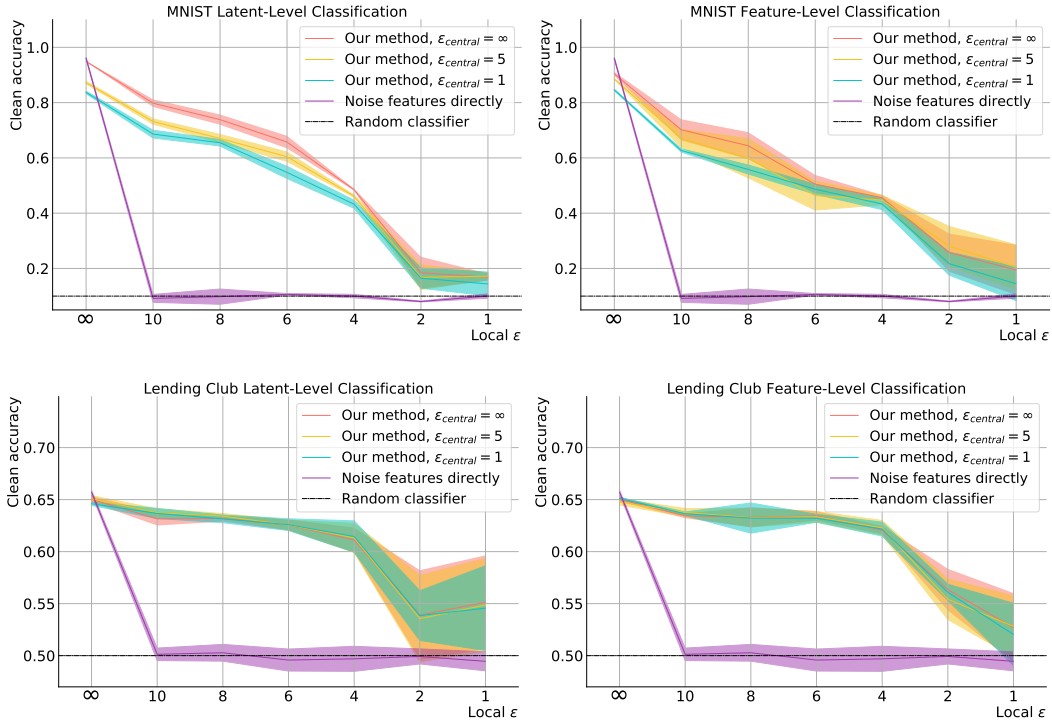

Figure 1: Clean accuracy as a function of local $\epsilon$ for data collection. Results are shown for the MNIST dataset (top) and Lending Club (bottom), on latent level (left) and feature level (right). Each line indicates a different value of $(\epsilon, \delta = 10^{-5})$-CDP at which the encoder was trained. The $x$-axis shows the $\epsilon$-LDP guarantee for the collected training set.

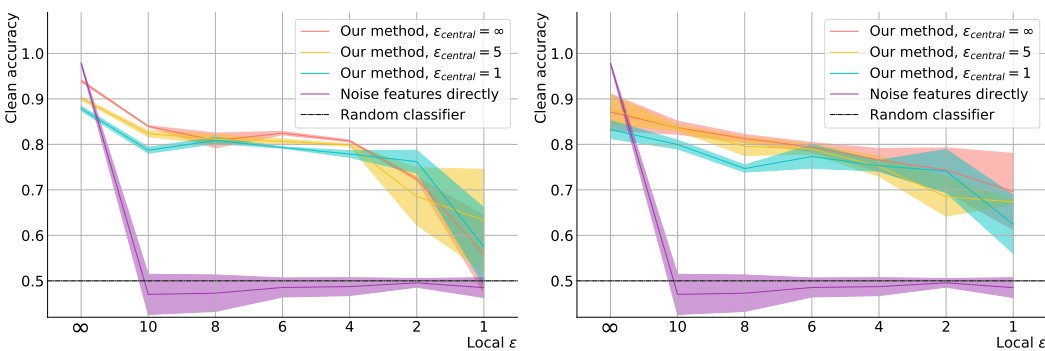

Figure 2: Clean accuracy as a function of local $\epsilon$ for novel-class classification on latent level (left) and feature level (right). Each line indicates a different value of $(\epsilon, \delta = 10^{-5})$-CDP at which the encoder was trained. The $x$-axis shows the $\epsilon$-LDP guarantee for the collected training set.

## 4.3 DATA JOINING

An organization training a classifier on some labelled dataset $\mathcal{D}_1$ could potentially improve performance by augmenting their dataset with other informative features, and so may want to join $\mathcal{D}_1$ with features from another dataset $\mathcal{D}_2$. We assume the owner of $\mathcal{D}_2$ may only be willing to share a privatized version of this dataset. For example, two organizations with mutual interests, such as the IRS and a private bank, or a fitness tracking company and a hospital, may want to join datasets to improve the performance of their algorithm. Similarly, it may be illegal for multinational organiza-

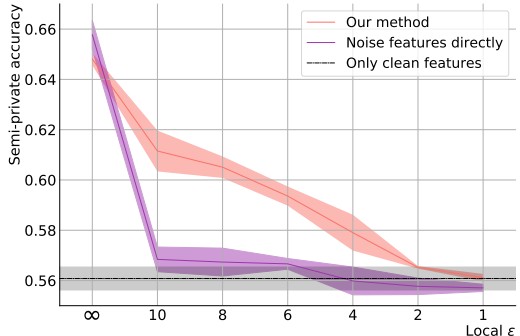 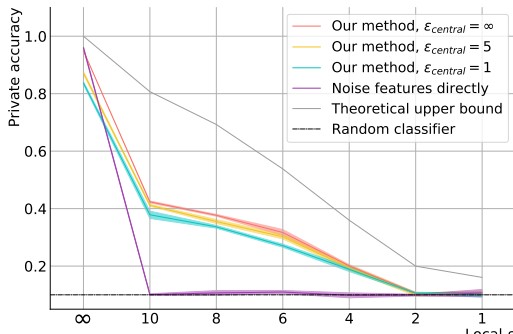

Figure 3: 'Semi-private' accuracy versus local $\epsilon$ for data joining (private features shared and joined on latent level). The $x$-axis shows the $\epsilon$-LDP guarantee for the collected training set.

Figure 4: Private accuracy versus local $\epsilon$ for latent-level data collection. Each line indicates a different value of $(\epsilon, \delta = 10^{-5})$-CDP at which the encoder was trained. The $x$-axis shows the $\epsilon$-LDP guarantee for the collected training set.

tions to share and join non-privatized client data between departments in different regions, but legal to do so when the shared data satisfies LDP.

We run this experiment on Lending Club, where we divide the dataset slightly differently from previous experiments: both datasets contain all rows, but $\mathcal{D}_1$ contains a subset of (clean) features, along with the clean label, and $\mathcal{D}_2$ contains the remaining features (to be privatized).

We follow a privatization procedure similar to that of Section 3.1, with the distinction that the VLM should be both trained on $\mathcal{D}_2$, and used to privatize $\mathcal{D}_2$. For the classification problem, instead of Equation 8, we optimize $\log p_\psi(y_1|x_1, \tilde{x}_2)$ where $(x_1, y_1) \in \mathcal{D}_1$ and $\tilde{x}_2 \in \mathcal{D}_2^{(\text{private})}$. We are not required to conduct a private grid search over hyperparmeters as in Section 3.3, since we have access to all raw data needed for validation. Note that unlike the previous two experiments, we train the classifier on a combination of both clean and privatized features, and we classify this same 'semi-private' group of features at inference time.

Results are shown in Figure 3. We can see that using features from $\mathcal{D}_1$ only, we obtain classification accuracy of 56.1%, while classifying on all (clean) features, we obtain 65.8% accuracy. The benchmark of noising the $\mathcal{D}_2$ features directly never achieves more than 1 percentage point accuracy increase over classifying the clean features only, whereas our model achieves a significant improvement for local $\epsilon \in [4, 10]$. We share the privatized features on latent level in this experiment and so do not need to satisfy CDP.

## 5 CLASSIFYING PRIVATE DATAPOINTS: APPLICATIONS AND EXPERIMENTS

In Sections 4.1 and 4.2, we have been investigating the use of privatized training data to train algorithms that classify clean datapoints. In some use cases however, we may want to train algorithms that act directly on LDP datapoints at inference time. Most notably, in the data collection framework, the organization may want to do inference on individuals whose data they have privately collected.

However from the definition of LDP in Equation 4, it is clear that a considerable amount of information about a datapoint $x$ is lost after privatization, and in fact, classification performance is fundamentally limited. In Appendix F, we show that for a given local $\epsilon$, the accuracy $A$ of a $K$-class latent-level classifier acting on a privatized datapoint (where the privatization mechanism has latent dimension $d \geq K/2$) is upper bounded by

$$A \leq \sum_{j=0, j \neq 1}^{K/2-1} \left( \binom{K/2-1}{j} \frac{(-1)^j}{1-j} \left[ \frac{e^{-j\epsilon/2}}{1+j} - \frac{e^{-\epsilon/2}}{2} \right] \right) - \frac{\epsilon+1}{8}(K-2) e^{-\epsilon/2} \quad (10)$$

In Figure 4, we show the accuracy of our model from Section 4.1 (data collection) when applied to privatized datapoints at inference time, and compare to the upper bound in Equation 10.

Running this experiment on MNIST, we see a considerable drop in performance when classifying privatized datapoints, compared with clean classification results.

We are clearly not saturating the bound from Equation 10. While it may initially appear that our model is under-performing, we note that our model aims to build a downstream-task-agnostic privatized representation of the data. This means that the representation must contain more information than just the class label. Meanwhile, the upper bound is derived from the extreme setting in which the latent encodes only class information, and would be unable to solve any other downstream task.

Though Equation 10 is constructed under the framework of latent-level classification, we do note that our feature-level classifier seems to marginally outperform the latent-level one (see Appendix G). This may be a result of the decoder de-noising the latent to some extent.

## 6 Conclusion and Future Work

In this paper we have introduced a framework for collecting and sharing data under the strict guarantees of LDP. We induce LDP by learning to efficiently add noise to any data type for which representation learning is possible. We have demonstrated a number of different applications of our framework, spanning important issues such as medical diagnosis, financial crime detection, and customer experience improvement, significantly outperforming existing baselines on each of these.

This is the first use of latent-variable models for learning LDP Laplace mechanisms. We foresee that even stronger performance could be achieved by combining our method with state-of-the-art latent-variable models that utilise more complex architectures, and often deep hierarchies of latents (Gulrajani et al., 2017; Maaløe et al., 2019; Ho et al., 2019). In this work we sought to show that significant hurdles in LDP for high-dimensional data could be overcome using a representation-learning driven randomization algorithm, which we indeed think is well established in this paper.

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

## A    PROOF THAT THE LOCAL LAPLACE MECHANISM SATISFIES LDP

**Claim:** The Local Laplace mechanism satisfies $\epsilon$-local differential privacy.

*Proof:*   We follow an approach similar to the proof in Dwork & Roth (2014) that the Laplace Mechanism satisfies CDP. Assume $x$ and $x'$ are two arbitrary datapoints.   Denote $\mathcal{M}^{\text{local}}(x) = f(x) + (s_1, \ldots, s_k)$ where $s_i \sim \text{Laplace}(0, \Delta f/\epsilon)$. Then for some arbitrary $c$ we know that

$$\frac{p(\mathcal{M}^{\text{local}}(x) = c)}{p(\mathcal{M}^{\text{local}}(x') = c)} = \prod_{i=1}^{k} \frac{p(\mathcal{M}_i^{\text{local}}(x) = c_i)}{p(\mathcal{M}_i^{\text{local}}(x') = c_i)} \tag{11}$$

$$= \prod_{i=1}^{k} \frac{\exp\left(-\frac{\epsilon|f_i(x)-c_i|}{\Delta f}\right)}{\exp\left(-\frac{\epsilon|f_i(x')-c_i|}{\Delta f}\right)} \tag{12}$$

$$= \prod_{i=1}^{k} \exp\left(\frac{\epsilon\big(|f_i(x')-c_i| - |f_i(x)-c_i|\big)}{\Delta f}\right) \tag{13}$$

$$\leq \prod_{i=1}^{k} \exp\left(\frac{\epsilon|f_i(x')-f_i(x)|}{\Delta f}\right) \tag{14}$$

$$= \exp\left(\frac{\epsilon||f(x')-f(x)||_1}{\Delta f}\right) \tag{15}$$

$$\leq \exp(\epsilon) \tag{16}$$

where the first inequality comes from the triangle inequality, and the second comes from the definition of $\Delta f$.

## B    PROOF THAT DECODED PRIVATE LATENTS SATISFY LDP

**Claim:** If a point in latent space satisfies $\epsilon$-LDP, then this point still satisfies $\epsilon$-LDP after being passed through a deterministic function, such as the function that parameterizes the mean of the decoder network.

*Proof:*   We follow an approach similar to the proof that central differential privacy is immune to post-processing (Dwork & Roth, 2014). Consider an arbitrary deterministic mapping $g : \mathcal{Z} \to \mathcal{X}$. Let $S \subseteq \mathcal{X}$ and $T = \{z \in \mathcal{Z} : g(z) \in S\}$. Then

$$p\big(g(\mathcal{A}(x)) \in S\big) = p\big(\mathcal{A}(x) \in T\big) \tag{17}$$

$$\leq \exp(\epsilon)\, p\big(\mathcal{A}(x') \in T\big) \tag{18}$$

$$= \exp(\epsilon)\, p\big(g(\mathcal{A}(x')) \in S\big) \tag{19}$$

## C    PROOF THAT THE FLIP MECHANISM SATISFIES LDP

**Claim:** The flip mechanism $p(\tilde{y} = i|y = j) = (1-p)\mathbb{I}(i = j) + p/(K-1)\mathbb{I}(i \neq j)$ where $p = (K-1)/(e^\epsilon + K - 1)$ satisfies $\epsilon$-local differential privacy.

*Proof:* We can write, for any $i, j, j'$:

$$\frac{p(\tilde{y} = i | y = j)}{p(\tilde{y} = i | y = j')} = \frac{(1-p)\mathbb{I}(i = j) + p/(K-1)\mathbb{I}(i \neq j)}{(1-p)\mathbb{I}(i = j') + p/(K-1)\mathbb{I}(i \neq j')} \tag{20}$$

$$= \begin{cases} \frac{(K-1)(1-p)}{p} & i = j, i \neq j' \\ \frac{p}{(K-1)(1-p)} & i \neq j, i = j' \\ 1 & \text{otherwise} \end{cases} \tag{21}$$

$$= \begin{cases} e^{\epsilon} & i = j, i \neq j' \\ e^{-\epsilon} & i \neq j, i = j' \\ 1 & \text{otherwise} \end{cases} \tag{22}$$

Therefore, we have that for any $i, j, j'$, $\frac{p(\tilde{y}=i|y=j)}{p(\tilde{y}=i|y=j')} \leq e^{\epsilon}$.

## D    IMPLEMENTATION REQUIREMENTS FOR DIFFERENT APPLICATIONS

In the scenario that $\mathcal{D}_1$ contains sensitive information, the encoder or decoder may need to be trained with the two-stage approach (outlined in Section 3.1) in order to guarantee $(\epsilon, \delta)$-CDP wrt $\mathcal{D}_1$. There are broadly two scenarios in which this is the case. Firstly, if private data is published on pixel level then a DP decoder is required. Secondly, if the encoder needs to be shared with the client (for example, in client side data collection) then a DP encoder is required. For pixel level data collection a DP decoder is not required, since the client can share their privatized latents and these can be 'decoded' to pixel level on the server side, avoiding the need to share the decoder. Table 1 explicitly outlines the CDP requirements for the applications discussed in our paper.

Table 1: Central differential privacy requirements for the VLM, with respect to the dataset $\mathcal{D}_1$.

| Application | Data Shared | Central-DP Requirement |
|---|---|---|
| Data Collection | Feature or latent level | DP-Encoder |
| Data Joining | Feature level
Latent level | DP-Decoder
None |
| Novel-Class Classification | Feature or latent level | DP-Encoder |

## E    EXPERIMENTAL SET UP

For every experiment in the paper, we conducted three trials, and calculate the mean and standard deviation of accuracy for each set of trials. The error bars represent one standard deviation above and below the mean.

We use the MNIST dataset and the Lending Club dataset. MNIST is a dataset containing 70,000 images of handwritten digits from 0-9 with corresponding class labels; the task is 10-way classification to determine the digit number. Lending Club is a tabular, financial dataset made up of around 540,000 entries with 23 continuous and categorical features (after pre-processing, before one-hot encoding); the task is binary classification, to determine whether a debt will be re-paid.

### E.1    DATA PRE-PROCESSING

For MNIST, we converted the images into values between 0 and 1 by dividing each pixel value by 255. These are then passed through a logit and treated as continuous.

For Lending Club, a number of standard pre-processing steps are performed, including:

- Dropping features that contain too many missing values or would not normally be available to a loan company.

- Mean imputation to fill remaining missing values.
- Standard scaling of continuous features. Extreme outliers (those with features more than 10 standard deviations from the mean) are removed here.
- Balancing the target classes by dropping the excess class 0 entries.
- One-hot encoding categorical variables.

The target variable denotes whether the loan has been charged off or not, resulting in a binary classification task. The train, validation, test split is done chronologically according to the feature 'issue date'.

In real world applications the sizes of the VLM training and validation sets, and the classifier training and validation sets would be pre-determined. For our experiments we used the data splits outlined in the following Sections.

### E.1.1 DATA COLLECTION

**MNIST:** The MNIST dataset contains 60,000 training points and 10,000 test points. We split both sets using 75% for the VLM and the remainder for the classifier. The VLM test set is used for validation since no test set is required here. The classifier training points are split randomly in a 9:1 ratio to form training and validation sets. We report classifier performance on the classifier test set.

**Lending Club:** This dataset is split into train, validation and test sets according to the issue date of the loans. The oldest 85% of data forms the training data, with the remaining forming the validation and test data. As with MNIST, we use 75% for the split between VLM training data and classifier training data.

### E.1.2 NOVEL-CLASS CLASSIFICATION

**MNIST:** We use a similar approach to the above, but split the data between the VLM and the classifier such that the VLM train/validation sets contain $\frac{8}{9}$ths of (unlabelled) training images from classes 0 to 8. The remaining $\frac{1}{9}$ths of 0 to 8 images, and all 9s, are used for classifier train, test and validation sets. Our VLM datasets then contain equal class balance for the classes 0 to 8, and the classifier datasets contain equal class balance for 9s and 'not 9s'.

### E.1.3 DATA JOIN

**Lending Club:** For this experiment, the datasets were split between the VLM and the classifier column-wise, between the dataset's 23 features. The VLM datasets contain 8 features (month of earliest credit line, number of open credit lines, initial listing status of loan, application type, address (US state), home ownership status, employment length, public record bankruptcies). The classifier datasets contain the remaining features. The feature split was chosen such that the first dataset contains some information to solve the classification task, but the features from the second dataset contain information which, at least before privatization, further improve classifier performance.

### E.2 NOISING FEATURES DIRECTLY

For continuous features, we assume that $\Delta f$ is equal to the difference between the maximum and minimum value of that feature within the training and validation sets used to train the VLM in the main experiments, after pre-processing. One then has to clip any values that lie outside this interval in the shared/collected dataset at privatization time.

### E.3 HYPERPARAMETER CHOICES

We conducted a grid search over a number of the hyperparameters in our model, in order to find the optimal experimental setup.

For stage 1 of the VLM training, a learning rate of $10^{-4}$ and batch size of 128 was used for Lending Club experiments, and a learning rate of $5 \times 10^{-4}$ and batch size of 64 was used for MNIST. We then searched over the following model hyperparameters (with central $\epsilon=\infty$):

Table 2: DP-Adam hyperparameters used for data collection and novel-class classification.

| Task | Central $\epsilon$ | Learning Rate | Batch Size | Noise Multiplier |
|------|------|------|------|------|
| MNIST | 5 | 5e-4 | 64 | 0.7 |
| | 1 | 5e-4 | 64 | 1.1 |
| Lending Club | 5 | 1e-4 | 128 | 0.56 |
| | 1 | 1e-4 | 128 | 1.1 |

Table 3: VLM hyperparameters used for data join experiments.

| Experiment | Task | Local $\epsilon$ | $d$ | $l$ | $\epsilon_{\text{pre-training}}$ |
|------|------|------|------|------|------|
| Data Joining | Lending Club | $\infty$ | 8 | 5 | 20 |
| | | 10 | 8 | 5 | 10 |
| | | 8 | 5 | 5 | 15 |
| | | 6 | 5 | 5 | 15 |
| | | 4 | 5 | 5 | 10 |
| | | 2 | 5 | 5 | 15 |
| | | 1 | 5 | 5 | 10 |

- The proportion $\lambda$ of our privacy budget assigned to the datapoint, compared with the label i.e. $\lambda = \epsilon_x/(\epsilon_x + \epsilon_y)$.

- The $\ell_1$ clipping distance $l$ of our inference network mean i.e. $l = \Delta\mu_\phi/2$.

- The Laplace distribution scale $b$ of our approximate posterior distribution during pre-training of the VLM. Note that we report this as the $\epsilon_{\text{pre-training}}$-LDP value induced by a sample from this posterior distribution, given $l$ in the previous point i.e. $\epsilon_{\text{pre-training}} = 2l/b$. This is fixed throughout training, unless 'learned' is specified in the below table, in which case the parameter $b$ is a learned scalar.

- The latent dimension $d$. This was fixed to 8 for data collection experiments but we searched over $d \in \{5, 8\}$ for the data join experiments due to the smaller number of features.

We also did a grid search over the following DP-Adam hyperparameters for central $\epsilon \in \{1, 5\}$:

- Noise multiplier
- Batch size
- DP learning rate

The DP-Adam hyperparameter max gradient norm was fixed to 1 throughout. The number of training epochs needed to reach the target central $\epsilon$ value follows from the choice of hyperparameters, combined with the VLM training set size (45,000 for MNIST, and 341,000 for Lending Club). Note that we fixed $\delta = 10^{-5}$ for all experiments.

The results from these grid searches are given in Tables 2, 3, and 4.

### E.4   NETWORK ARCHITECTURES

Throughout the paper, we used feedforward architectures for both the VLM and classifier networks.

For MNIST, we use a VLM encoder network with 3 hidden layers of size $\{400, 150, 50\}$, and a decoder network with 3 hidden layers of size $\{50, 150, 400\}$. For Lending Club, we use a VLM encoder and decoder network with 2 hidden layers of size $\{500, 500\}$. For the the latent level classifier we used a network with 1 hidden layer of size 50 and for pixel level classifier we use a network with 3 hidden layers of size $\{400, 150, 50\}$.

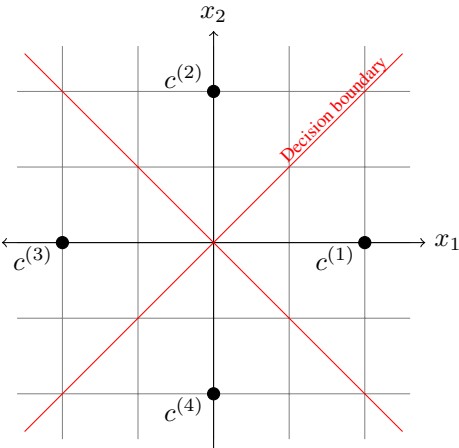

Figure 5: The decision boundary for a classifier that equally separates (in $\ell_1$-distance) vertices $c^{(i)}$ for $i \in \{1, 2, 3, 4\}$ in 2-dimensional space.

## F  PROOF OF UPPER BOUND ON NOISY ACCURACY

In this section, we derive an upper bound on accuracy for the classification of datapoints privatized using the Laplace mechanism (see Equation 4). To simplify the proof, we make the following assumptions:

- We have $K$ equally balanced classes.

- $K$ is even.

- $K \leq 2d$ where $d$ is the dimension of the output of $f(\cdot)$ (the latent space on which we add Laplace noise).

These are true for all experiments in this paper.

First, suppose that $K = 2d$. Since we add iid Laplace noise to each datapoint, we will obtain the highest possible accuracy when $f(\cdot)$ maps datapoints from class $i$ as far away from datapoints from class $j \neq i$ as possible. This maximum separation distance can be at most $\Delta f$; we can separate all $K$ classes by distance $\Delta f$ in our $d$ dimensional latent space iff each class is mapped to a separate vertex $c^{(y)}$ of the taxicab sphere of ($\ell_1$-norm) radius $\Delta f / 2$.

The decision boundary is given by the line that equally separates these vertices in $\ell_1$-space, as shown (for 2 dimensions) in Figure 5.

The accuracy of the classifier $\mathcal{C}$ on datapoints privatized by the latent-space Laplace mechanism $q(\cdot|x) \sim \mathrm{Laplace}(f(x), \Delta f / \epsilon)$ is given by

$$A = \mathbb{E}_{(x,y) \sim p(x,y), \tilde{z} \sim q(z|x)} p\left(\mathcal{C}(\tilde{z}) = y\right) \tag{23}$$

$$= \mathbb{E}_{y \sim p(y)} p\left(\mathcal{C}(c^{(y)} + s) = y\right) \tag{24}$$

$$= p\left(\mathcal{C}(c^{(1)} + s) = 1\right) \tag{25}$$

where $s = (s_1, \ldots, s_d)$ and $s_i \sim \mathrm{Laplace}(0, \Delta f / \epsilon)$. The second equality follows from the fact that all points from a given class are mapped to the same point in latent space, and the final equality follows from the symmetry between classes. This final term gives the probability that when we add Laplace noise to $c^{(1)}$ and obtain the private representation $\tilde{c}^{(1)}$, we do not cross the decision boundary. We assume WLOG that $c^{(1)} = (1, 0, \ldots, 0)$ and calculate this probability as follows (dropping the superscript for clarity)

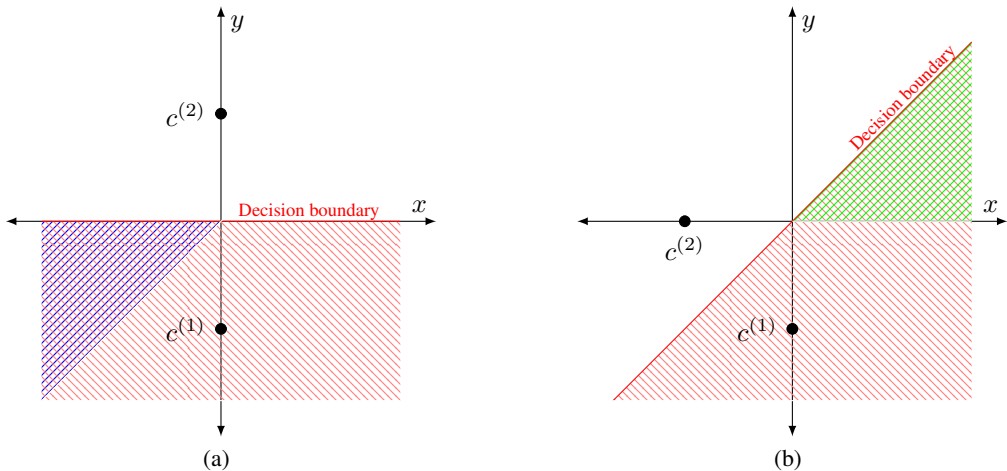

Figure 6: The shaded areas represent, for $d = 2$ and $K = 2$, the decision boundaries for: (a) a function $f(\cdot)$ that maps the two classes onto opposing vertices; (b) a function $f(\cdot)$ that maps the two classes onto adjacent vertices. Refer to Appendix F for details on the color-coding of shaded areas.

$$A = \int_{\tilde{c}_1 > 0, \ \tilde{c}_i < |\tilde{c}_1|, \forall i \neq 1} d\tilde{c}_1 \ldots d\tilde{c}_d \frac{1}{(2b)^d} \exp\left( - \frac{||\tilde{c} - c||_1}{b} \right) \tag{26}$$

$$= \int_0^\infty d\tilde{c}_1 \frac{1}{2b} \exp\left( - \frac{|\tilde{c}_1 - 1|}{b} \right) \prod_{i=2}^d \int_{-\tilde{c}_1}^{\tilde{c}_1} d\tilde{c}_i \frac{1}{2b} \exp\left( - \frac{|\tilde{c}_i|}{b} \right) \tag{27}$$

$$= \int_0^\infty d\tilde{c}_1 \frac{1}{2b} \exp\left( - \frac{|\tilde{c}_1 - 1|}{b} \right) \left( 1 - \exp(-\tilde{c}_1/b) \right)^{d-1} \tag{28}$$

$$= \int_0^\infty d\tilde{c}_1 \frac{1}{2b} \exp\left( - \frac{|\tilde{c}_1 - 1|}{b} \right) \sum_{j=0}^{d-1} \binom{d-1}{j} (-1)^j \exp\left( - \frac{\tilde{c}_1}{b} \right)^j \tag{29}$$

$$= \sum_{j=0}^{d-1} \binom{d-1}{j} (-1)^j \int_0^\infty d\tilde{c}_1 \frac{1}{2b} \exp\left( - \frac{|\tilde{c}_1 - 1|}{b} \right) \exp\left( - \frac{j\tilde{c}_1}{b} \right) \tag{30}$$

$$= \sum_{j=0}^{d-1} \binom{d-1}{j} (-1)^j \left[ \int_0^1 d\tilde{c}_1 \frac{1}{2b} \exp\left( - \frac{\tilde{c}_1(j-1) + 1}{b} \right) \right.$$
$$\left. + \int_1^\infty d\tilde{c}_1 \frac{1}{2b} \exp\left( - \frac{\tilde{c}_1(j+1) - 1}{b} \right) \right] \tag{31}$$

$$= \frac{1-d}{2b} \left( 1 + \frac{b}{2} \right) \exp(-1/b) + \sum_{j=0, j \neq 1}^{d-1} \binom{d-1}{j} (-1)^j \left[ \frac{\exp(-j/b)}{1-j^2} - \frac{\exp(-1/b)}{2(1-j)} \right] \tag{32}$$

$$= (1-d) \frac{\epsilon + 1}{4} \exp(-\epsilon/2) + \sum_{j=0, j \neq 1}^{d-1} \binom{d-1}{j} (-1)^j \left[ \frac{\exp(-j\epsilon/2)}{1-j^2} - \frac{\exp(-\epsilon/2)}{2(1-j)} \right] \tag{33}$$

where in the last step we used the fact that in this case $\epsilon = 2/b$. By substituting $d = K/2$, Equation 10 follows directly.

Now, we consider the case $K \leq 2d$. Clearly, the taxicab sphere has more than $K$ vertices, and so classes can occupy different combinations of vertices. The maximum accuracy will be found where the occupied vertices are maximally separated from each other. For the case of Laplace noise, where probability mass decreases exponentially with $\ell_1$-distance from the mean, it is clear

that the probability of a noised datapoint crossing a decision boundary is higher when the classes are centered on vertices aligned along different axes. We have shown this for $d = 2$ in Figures 6(a) and 6(b), where clearly the probability mass of the blue shaded area for a Laplace distribution with mean $c^{(1)}$ is larger than the green shaded area. Therefore we are more likely to cross the decision boundary in Figure 6(b), given a fixed quantity of noise.

Thus, the optimal setup is when the $K$ classes are positioned on vertices aligned along the first $K/2$ axes. Noise added to the remaining $d - K/2$ dimensions does not affect the classifier, and so Equation 33 still holds.

## G  FEATURE LEVEL PRIVATE CLASSIFIER FOR DATA COLLECTION

Figure 7 shows the private accuracy results for MNIST data collection, on latent level.

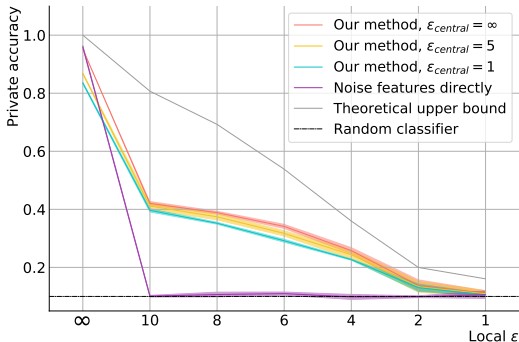

Figure 7: Private accuracy as a function of local $\epsilon$ for data collection (feature level). Each line indicates a different value of $(\epsilon, \delta = 10^{-5})$-CDP at which the encoder was trained, each point on the $x$-axis shows the $\epsilon$-LDP guarantee for the collected training set.

Table 4: VLM hyperparameters used for data collection and novel-class classification.

| Experiment | Task | Local $\epsilon$ | $\lambda$ | $l$ | $\epsilon_{\text{pre-training}}$ |
|---|---|---|---|---|---|
| Clean Accuracy, Latent Level Classification | MNIST | $\infty$ | N/A | 10 | learned |
| | | 10 | 0.7 | 10 | 27 |
| | | 8 | 0.7 | 5 | 29 |
| | | 6 | 0.7 | 5 | 15 |
| | | 4 | 0.7 | 7.5 | 5 |
| | | 2 | 0.7 | 7.5 | 21 |
| | | 1 | 0.7 | 5 | 15 |
| | Lending Club | $\infty$ | N/A | 10 | learned |
| | | 10 | 0.7 | 5 | 15 |
| | | 8 | 0.7 | 5. | 29 |
| | | 6 | 0.7 | 5. | 29 |
| | | 4 | 0.7 | 5 | 15 |
| | | 2 | 0.95 | 5 | 15 |
| | | 1 | 0.95 | 10 | 21 |
| Private Accuracy, Latent Level Classification | MNIST | $\infty$ | N/A | 10 | learned |
| | | 10 | 0.7 | 10 | 5 |
| | | 8 | 0.7 | 7.5 | 5 |
| | | 6 | 0.7 | 7.5 | 5 |
| | | 4 | 0.7 | 5 | 5 |
| | | 2 | 0.7 | 5 | 15 |
| | | 1 | 0.7 | 7.5 | 15 |
| | Lending Club | $\infty$ | N/A | 10 | learned |
| | | 10 | 0.95 | 5 | 15 |
| | | 8 | 0.95 | 5. | 15 |
| | | 6 | 0.7 | 5. | 15 |
| | | 4 | 0.7 | 5 | 15 |
| | | 2 | 0.95 | 5 | 29 |
| | | 1 | 0.95 | 10 | 21 |
| Clean Accuracy, Feature Level Classification | MNIST | $\infty$ | N/A | 10 | learned |
| | | 10 | 0.7 | 5 | 29 |
| | | 8 | 0.7 | 5 | 15 |
| | | 6 | 0.7 | 7.5 | 21 |
| | | 4 | 0.7 | 5 | 5 |
| | | 2 | 0.7 | 7.5 | 5 |
| | | 1 | 0.7 | 5 | 15 |
| | Lending Club | $\infty$ | N/A | 10 | learned |
| | | 10 | 0.7 | 5 | 29 |
| | | 8 | 0.95 | 5. | 29 |
| | | 6 | 0.7 | 5. | 15 |
| | | 4 | 0.7 | 5 | 15 |
| | | 2 | 0.7 | 5 | 15 |
| | | 1 | 0.7 | 10 | 15 |
| Private Accuracy, Feature Level Classification | MNIST | $\infty$ | N/A | 10 | learned |
| | | 10 | 0.7 | 10 | 5 |
| | | 8 | 0.7 | 10 | 5 |
| | | 6 | 0.7 | 10 | 5 |
| | | 4 | 0.7 | 5 | 5 |
| | | 2 | 0.7 | 7.5 | 5 |
| | | 1 | 0.95 | 5 | 15 |
| | Lending Club | $\infty$ | N/A | 10 | learned |
| | | 10 | 0.95 | 5 | 15 |
| | | 8 | 0.95 | 5. | 15 |
| | | 6 | 0.95 | 5. | 15 |
| | | 4 | 0.7 | 5 | 15 |
| | | 2 | 0.95 | 7.5 | 27 |
| | | 1 | 0.95 | 5 | 15 |

