# OpenReview forum: "Learning to Noise: Application-Agnostic Data Sharing with Local Differential Privacy"
_ICLR.cc/2021/Conference — Reject_

### Official Review · AnonReviewer4 · 2020-10-26
**A well-written paper on an important subject. Good results, needs some improvements.**

**Rating:** 6
**Confidence:** 3

**Review:**

Summary:
This paper presents a new privatization mechanism for Local Differential Privacy based on representation learning. The proposed VAE-based method is used for the low-dimensional latent representation of the data and uses the Laplace mechanism to satisfy Local DP. The paper shows this mechanism can be used across various applications such as private data collection, private novel-class classification, data joining, etc.

The paper is clear and easy to follow. The proposed method provides a great solution for data sharing with the local DP and can be used in many real-world applications. However, some clarification is needed. The experimental results can be improved by adding more baselines. Some important/recent references are also missing.

Major Comments:
-         It would be better if the authors add a related work section or extend the literature review paragraph in Introduction and include more recent work in this area and point out how the proposed work advances the state-of-the-art. There are several works on DP for high-dimensional data such as:
AutoGAN-based Dimension Reduction for Privacy Preservation by Nguyena et al.
P3GM: Private High-Dimensional Data Release via Privacy Preserving Phased Generative Model by Takagi et al.
There are also several existing works on LDP based on VAE. The authors are expected to state the differences between the existing work and the proposed work.
-         The authors mentioned the DP Synthetic data models need large data, this is also the case for training the VAE in this work. Also, they mentioned these techniques need labeled data. DP-GAN models need access to real data for training (no label is required) and then it can be used indefinitely for generating synthetic data. Please clarify this.
-         In an existing work (P3GM paper mentioned above), it is shown that VAE’s objective function is too sensitive to the noise of DP-SGD, how the authors tackle this problem? And how it affects the final results?
-         The baseline methods are limited to "direct noise features" only. It would be better if the authors use other techniques such as some recent work on LDP on high dimensional data or other general DP classifiers, DP-SGD, PATE, etc. as the baselines for this experiment.
-         It would be better if the authors also showed the performance of the proposed method on higher dimensional data (e.g., Lending club has only 23 features)


Minor comments:
-         In the introduction, please list the contribution of this work.
-         Please define all variables in Eq 5-6.
-         Please add more up-to-date references.

---

> ### Author Response · Authors · 2020-11-14
> **Literature review extended; benchmarks justified**
>
> We thank the reviewer for their response, and have addressed each of their points in order below.
>
> **Major Point 1:** We have updated the introduction to include a more in depth discussion of the related work, including the papers cited in this review.
>
> **Major Point 2:** Thank you for pointing this out, we have updated this literature review to reflect the fact that not all work in synthetic generation requires labels for training, and more importantly to outline how our work differs from existing work in the literature.
>
> In short, our method is solving a very different problem to the work on synthetic generation, and we believe synthetic generation to be limited in its applications. As mentioned in our response to AnonReviewer1 point 4, we learn an LDP mechanism for privatizing individual datapoints or subsets of features, for use in downstream tasks; this opens up possibilities to solve a much broader range of problems. Synthetic generation approaches create datasets that are not related to individuals, but are simply likely under the learnt probability distribution. One has no way of collecting / privatizing new data with these approaches, as is required for the tasks in Sections 4.1, 4.2 or 5.
>
> **Major Point 3:** In P3GM, it is suggested that training a VAE end to end with DP-SGD is challenging, and this is something we also found. We found that the simple 2-stage approach outlined at the end of Section 3.1 was sufficient for learning good representations. P3GM also adopts a technique whereby the training of the VAE is split into steps, however their approach seems more complicated than ours. It should be stated that we are attempting to solve a completely different problem to P3GM as described in our rebuttal to Major Point 2 above.
>
> **Major Point 4:** As discussed in our response to AnonReviewer1 point 4, we do not see an obvious way to benchmark against DP classifiers due the nature of the tasks solved by our model in this paper. The literature on local differential privacy for high dimensional data is still in its infancy, and much of the existing work is tailored to specific datasets rather than a general model such as ours. For example [1, 2] focus on the collection of data over time, whilst [2] note that for one time collection “direct randomization on the true client’s value is sufficient to provide strong privacy protection”. Indeed this direct randomization approach is in line with our benchmark. These papers have been discussed in the literature review within Section 1.
>
> **Major Point 5:** We tested our algorithm on MNIST, which contains 784 features, and do not consider this to be a low-dimensional problem. Even if, in the context of SOTA non-DP computer vision work, one might categorise MNIST as low dimensional, this is certainly not the case in the context of LDP research. By applying the algorithm to both MNIST and Lending Club, we demonstrate the versatility of the approach. It should theoretically work on any data type for which you can train a VAE to learn good representations. This allows us to make use of a vast body of work that has already been achieved in the generative modelling community.
>
> **Minor Points:** All three minor points have been addressed in the updated manuscript.
>
> We hope that these clarifications as well as the updates to our manuscript have addressed the reviewer's comments, and that the reviewer will consider revising their score accordingly.
>
> [1] Ding et al. - Collecting telemetry data privately, 2017
> [2] Erlingsson et al. -  RAPPOR: Randomized aggregatable privacy-preservingordinal response, 2014.

---

> > ### Comment · AnonReviewer4 · 2020-11-23
> > **More comments and concerns**
> >
> > The authors have extended the literature review and addressed some of my concerns. However, it seems the applications of the proposed work are limited to the cases where the public data is available for learning a VAE. This assumption significantly limits the work for many real-world applications where such data is not available. The authors are expected to emphasize this in the paper including the abstract.
> >
> > Also, I still believe the benchmarking should be improved at least by comparing to some LDP methods such as LoPub by Ren et al. If not possible, the following should be updated: "We achieve significant gains in performance on downstream classification tasks relative to benchmarks that noise the data directly, which are state-of-the-art in the context of application-agnostic LDP mechanisms for high-dimensional data"

---

> > > ### Author Response · Authors · 2020-11-24
> > > **Benchmark claims updated; Clarification of data requirements;**
> > >
> > > The VLM requires a clean, unlabelled training dataset that follows a similar distribution to the dataset one wishes to share under LDP. In many cases this would be a dataset that the organisation already has access to, rather than a public dataset. For example, it is highly likely that a public health body would have access to some kind of proprietary medical imaging dataset, or that a technology company would already have collected some data from a group of users (e.g. in jurisdictions where data privacy laws are weaker). In the scenario that sensitive data is used, one would use DP-Adam to train the VLM and protect the members of this clean dataset. Indeed, DP-Adam was used in all our experiments to reflect this.
> > >
> > > Furthermore, we have outlined several real-world applications throughout Sections 4 and 5, as well as two explicit examples in our latest response to AnonReviewer1, which we believe are compelling and particularly relevant to real-world scenarios (across healthcare, finance, and consumer industries).
> > >
> > > Regarding benchmarks, Ren et al. state: “our goal is to help the central server publish a synthetic dataset that has the approximate joint distribution of d attributes with local privacy”. We stress that synthetic dataset generation is an inherently different task, and such models cannot solve many of the problems described in this paper.
> > >
> > > For example, the data join problem outlined in Section 4.3 requires that the privatized data be associated with specific individuals in order to be joined with another database. With LoPub, the data is a collection of synthetic random samples from a similar distribution, which do not represent individuals. The data join task is therefore not possible. Similarly, no evidence is provided in Ren et al. that LoPub is able to classify directly on its collected, privatized data, and so the experiments in Section 5 are also out of the scope of this model. Benchmarking against LoPub would require significant adaptations to their proposed model, which to the best of our understanding, would not be in line with the authors’ original motivation. That being said, we have updated the sentence specified to reflect the fact that our approach tackles problems in data sharing that are not possible with existing methods.

---

### Official Review · AnonReviewer2 · 2020-10-27

**Rating:** 6
**Confidence:** 2

**Review:**

For LDP, when applying noise directly to high-dimensional data, the required noise entirely destroys data utility. In this paper, authors introduce a novel, application-agnostic privatization mechanism that leverages representation learning to overcome the prohibitive noise requirements of direct methods. They further demonstrate that this privatization mechanism can be used to train machine learning algorithms across a range of applications. They achieve significant gains in performance for high-dimensional data.

In this paper, authors have benchmarked results against such a mechanism, in which add Laplace noise to all continuous features, and flip each of the categorical features with some probability.
For high-dimensional datasets, features are often highly correlated; consequently, noising features independently is wasteful towards privatizing the information content in each datapoint. A more effective approach to privatization involves noising a learned lower-dimensional representation of each datapoint using a generic noising mechanism. Applying the Laplace mechanism thus ensures the encoded latents, as well as reconstructed datapoints satisfy LDP. They focus on classification tasks. At inference time, they show that this classififier can act on either clean or privatized datapoints,.

For the writing, it’s better to give a clear algorithm. For the experiment, when epsilon<=10, the accuracy is not very good. The related work and comparisons are not enough. They are quite a few work about LDP learning can be literature reviewed. By the way, we usually use private data rather than privatized data.

---

> ### Author Response · Authors · 2020-11-14
> **Related work extended; performance justified**
>
> Thank you for the valuable feedback; we hope that points below adequately address your concerns.
>
> Re: “it’s better to give a clear algorithm”
>
> AnonReviewer3 also sought clarification in this regard. We agree this could have been clearer and have significantly edited Section 3.1 to clarify the privatization procedure. Please let us know if you still believe it is unclear.
>
> Re: “the accuracy is not very good”
>
> LDP is an extremely strict criterion, and training models under LDP involves a privacy-utility trade off. Equation 9 and Figure 4 outline a strict upper bound on what the model can achieve, and demonstrate how one cannot directly compare an LDP model against a classifier trained without privacy guarantees. Even in comparison to models trained with CDP, such as those from (Abadi et al. 2016), the LDP criterion is incredibly strict, as it protects the privacy of individual datapoints. Very little work has been done on LDP mechanisms for high dimensional data.
>
> Many applications (including those that were studied in our paper) are impossible without LDP. Therefore, our results should be compared against the best technique available to the application, rather than the standard techniques that we are used to in less private applications.
>
> Re: “The related work and comparisons are not enough”
>
> We have extended our literature review to give a greater overview of this research in this field.

---

### Official Review · AnonReviewer1 · 2020-10-29
**Well written but lack of theoretical discussions and weak empirical studies.**

**Rating:** 3
**Confidence:** 4

**Review:**

In this paper, the authors present a generative-model-based Laplace mechanism. By training the VAE on some dataset, the trained encoder can be used to privatize raw data towards epsilon, delta-LDP. Though the method is novel, the privacy guarantee of the proposed method is not clearly stated and proved. Related experiments are not convincing, either.

**Strength**
The paper is well written with a clear motivation, explanation of methodology. To my knowledge, I believe the work is useful for the privacy research community. The proposed method is also novel.

**Weakness**
- The motivation to use the Laplace mechanism is not very clear. At the beginning of Sec. 2, the authors reason the usage by "as it provides strong theoretical privacy guarantees". This is not convincing for readers especially for those who are not familiar with LDP. Since the Laplace mechanism directly comes from the CDP, I would wonder how does the Gaussian mechanism works. How does the Laplace mechanism guarantee privacy better than the Gaussian mechanism? Reference or proof is essential here.

- In page 3, the authors briefly mention that the local version of the Laplace mechanism can be epsilon-LDP if the sensitivity is accordingly defined. This really lacks rigorousness. In the following sections, the authors refer to (Dwork and Roth, 2014) for the post-processing theorem. Since the work (Dwork and Roth, 2014) is mainly about CDP, I am not sure how the post-processing theorem can be adopted for LDP. Either reference or clear proof is required.

- Meanwhile, there lacks an end-to-end proof of the privacy guarantee of the VAE. I am not sure if the proposed VLM training guarantees privacy. Either, the privacy of encoding is not very clear. Especially, there involves a non-private training on stage 1.

- The experiments are run with pretty week baselines. Through this paper, the authors actively use the same conclusion from CDP (Dwork and Roth, 2014). Thus, I suppose the state-of-the-art CDP algorithms should also be applicable to the experimented tasks, e.g, classification. For the specific task, how well is the proposed compared to the SOTA CDP private learning algorithms? For example, (Abadi, et al., 2016), or (Phan, et al. 2017). Especially, (Phan, et al. 2017) also proposed an adaptive Laplace mechanism without depending on pre-training of the mechanism.

- In page 4, the DP-Adam mentioned in Stage 2 is not stated or proved in (Abadi et al., 2016). Only DP-SGD was discussed. A strict proof is required for the DP-Adam which intensively re-uses private results to help improve the gradients. Thus, the privacy guarantee is not straightforward.

- Seems the VLM training is using a non-DP optimizer at stage 1. Then how the whole training could guarantee privacy on the VLM training set. In experiments, the VLM training set is directly extracted from the private dataset (MNIST). Even though the author experiments with diverse D_1 D_2 distribution for VLM train/test in Sec 4.2, the two datasets are still from the same dataset. In practice, when such a D_2 is private, it is hard to find a D_1 to be non-private. I am afraid this could cause serious privacy leakage. Therefore, I doubt if the experimental results are useful for proving the effectiveness of a private algorithm. More realistic dates should be used.

- In Sec 4.1, the authors run the experiments in two steps. First, the VLM is trained with 'a DP encoder using D_1'. I am not clear how the DP encoder comes from. Does the VLM is also trained with DP? The setting has to be clarified.

- The experiment comparison seems not fair for baselines. For VLM, there are two datasets for training VLM and encoding classification train data. However, the baseline only has classification training data. The VLM encoder has additional information about the data distribution or the noise (by back-propagation in VLM training). The unfairness in the information could be the core reason for the difference in performance. How does the baseline perform if it is pre-trained and tuned (hyper-parameters) on another dataset?

(Phan, et al. 2017). Adaptive Laplace Mechanism: Differential Privacy Preservation in Deep Learning

---

> ### Author Response · Authors · 2020-11-14
> **Privacy guarantees clarified; sufficiency of empirics justified**
>
> We thank the reviewer for their detailed feedback. Point-by-point responses are provided below:
>
> **Points 1 & 2:** We chose the Local Laplace mechanism as it guarantees (epsilon, delta=0)-LDP, while the Gaussian mechanism guarantees (epsilon, delta>0)-LDP which is not as strict. We believe that the method should work using the Gaussian mechanism; this would require a minimal change to the model (changing the prior and approximate posterior to be Gaussian). A paragraph has been added to Section 2 to discuss the choice of Laplace versus Gaussian mechanisms. We have also added a formal definition of the local Laplace mechanism in Section 2, and a proof that this guarantees LDP in Appendix A. We have added a proof that LDP is immune to post-processing in Appendix B.
>
> **Points 3 & 7:** There are two separate types of data in our approach: data which is transferred between entities and data which is only used for training the VLM. The latter must satisfy CDP so that the VLM parameters can be shared, while the former must satisfy LDP so that it can be safely shared. Because these two requirements are completely independent, keeping track of privacy is relatively straightforward:
> * DP training (e.g. DP-Adam) must be used when training the parameters of the VLM that are to be shared. This is done in stage 2 of VLM training as outlined in Section 3.1.
> * LDP is required for transferring the data. This is achieved when the VLM adds noise in the latent space.
> With this clarification of the two-stage approach, and the proofs that have now been added to Appendix A and B, the LDP and CDP guarantees are proved.
>
> **Point 4:** Our approach aims to tackle problems in which data must be privately shared between entities, thus requiring a LDP mechanism. The experiments in our paper require data sharing, and thus CDP approaches such as those referenced, cannot be used to tackle this problem. Consequently, they cannot be applied as benchmarks in our work.
>
> **Point 5:** Thank you for pointing this out. We had missed a citation for [Gylberth et al. - Differentially Private Optimization Algorithms for Deep Neural Networks] who show that DP-Adam satisfies CDP.
>
> **Point 6:** With respect to privacy leakage, we have clarified this concern in our response to points 3 & 7 above. The reviewer also comments on the relevance of our data set choices. We describe in Sections 4.1-4.3 and Section 5 multiple different scenarios in which the dataset assumptions we have made are highly relevant.
>
> **Point 8:** We agree that with access to a similar, pre-training dataset $D_1$, we are at an advantage over our baseline, and this is the motivation for the paper. We are demonstrating that, with some knowledge about the structure of our data distribution, one can learn a mechanism to privatise the data which is more effective than a fixed mechanism. This setup is a common scenario for organisations looking to privately exploit data, as we have articulated in our applications.
>
> In summary, we have made significant improvements to the manuscript in response to some of the reviewer's comments (e.g. extensive detail added to address the privacy guarantees), and hopefully provided helpful clarification in cases where perhaps the reviewer misunderstood our work (e.g. the applicability of central DP). With this, we hope the reviewer will reconsider their score.

---

> > ### Comment · AnonReviewer1 · 2020-11-23
> > **A few more questions**
> >
> > On Point 4: Although these methods are not designed for private data sharing, they tackle the same problem, private classification learning. When these methods can privately train classifiers well, why do we need to share private data?
> >
> > On Point 8:
> > 1. Can you elaborate on what knowledge about the data distribution is essential for the training? For example, to what extent the similarity is the proposed method can work well?
> > 2. It will be necessary that the pre-training dataset is really public. Extracting a sub-set from a so-called private dataset is not a convincing way. In practice, we are unable to split part of the private dataset for pre-training only given a private dataset.
> > 3. I cannot agree that the method is practically better than the baseline if the public data are only used by the proposed method. The improper setting in experiments will fail to reveal the inherent reason why the method work.

---

> > > ### Author Response · Authors · 2020-11-23
> > > **Dataset availability clarified; Applications explained and model comparisons justified**
> > >
> > > __Point 4:__ This work is proposing a solution to the task of private data collection / sharing under LDP. In our experiments, we aim to demonstrate that the privatized data retain sufficient information for training ML algorithms; we do this by training a downstream classifier on the collected dataset but emphasise that we could use the privatized data for a range of other tasks. The goal of this paper is to solve the problem of private data sharing, which is not only applicable in situations where training a private classifier is not an option, but also provides a much greater flexibility to the data collector.
> > >
> > > To concretely outline an application where data sharing is relevant (and training a private classifier would not be possible), consider the data join problem in Section 4.3. Suppose a tax authority wants to investigate an individual who has an account with a private bank. Transaction details from the bank may be useful in the investigation. Thus the private bank could train a VLM on their transaction dataset, and this could be used to privatize the transactions of the individual, before sending to the authorities. The authorities could then join these privatized features with the clean features they already have on the individual. Our experiments demonstrate that by joining private and clean features, we have more information about the individual than with only clean features.
> > >
> > > Importantly, in this example, the tax authority will want to do many things with this new joined data set. They may want to train a classifier, but they will also need to be able to audit their work with access to the data used, among many other things. Therefore, a privately trained classifier is not relevant for this application.
> > >
> > > __Point 8:__ As we understand, the reviewer is concerned that the availability of datasets used in our work is unrealistic. We address this point below, but please clarify if we have misunderstood the reviewer’s position.
> > >
> > > There are two datasets in our approach:
> > >
> > > 1. The pre-training dataset is a clean (i.e. not privatized) dataset used to train the VLM. If this is a sensitive dataset, and not publicly available, the organisation collecting the data would use DP-Adam to protect the members of this dataset.
> > >
> > > 2. The second dataset is a sensitive dataset to be collected by the organisation. The VLM is given to the data owners, who privatize their data locally before sending it to the organisation.
> > >
> > > Our work crucially provides a framework that allows data owners (who are apprehensive about sharing sensitive data) to share this second dataset with an organisation, under LDP.
> > >
> > > Throughout Sections 4 and 5, we describe numerous contexts in which it is realistic that an organisation would have access to a pre-training set, as in our experiments. For clarity, we explicitly outline one application here, which relates to the experiment in Section 4.2. Suppose a public health body wishes to collect chest scans from different hospitals for patients with a novel disease. The health body would have access to historic chest scan datasets on which they could train a VLM. The VLM would then be sent to each hospital, and chest scans of patients with the novel disease would be privatized locally and sent back to the health body, forming a privatized dataset. This could then be used in many downstream tasks, one of which is to train a novel disease classifier.
> > >
> > > We hope this example has sufficiently demonstrated one example of the type of situation in which our data assumptions are realistic. We also hope that this example reinforces our response to point 4, in highlighting another application of our work in which private classifiers cannot be used.

---

### Official Review · AnonReviewer3 · 2020-10-30
**This work proposes an application-agnostic way to generate LDP representations of sensitive data or synthetic data that satisfies LDP. The proposed approach is effective for high-dimensional data. Downstream ML tasks can take these representations or synthetic data without worrying about privacy leakage, and achieve better accuracy than existing LDP solutions**

**Rating:** 6
**Confidence:** 5

**Review:**

Strong point 1: The idea of putting noise insertion (via noisy data-generation models) and optimization of good representations together to obtain LDP representations and/or synthetic data seems to be effective. While (6) relies on some independency assumptions, it might be fine in most cases and empirical evidence is reported to support it

Strong point 2: It is an application-agnostic approach and theoretically any downstream tasks and models can be supported... When there is a label, the privacy budget is split and random perturbation is used on labels

Strong point 3: It outperforms naive LDP baselines (with noise added directly to features) a lot in experiments

Weak point 1: The proof of the most important result is missing: It is said that "sampling from $q_\phi(z|x)$ produces a representation $\tilde z$ of $x$ that satisfies $\epsilon$-LDP. I don't think it is a trivial result and the author needs to everything together (including the analysis of sensitivity, the optimization algorithm, and so on) to formally prove it

Weak point 2: A minor issue: in figures of experiments, by "clean accuracy", do you actually mean "accuracy" (for some algorithms in the figures, it is privacy accuracy?)

W1 is the main reason for the rating of 6 but not higher ones - highly encourage the authors to fix it before the publication

---

> ### Author Response · Authors · 2020-11-14
> **Requisite privacy-guarantee proofs added**
>
> We thank the reviewer for their valuable feedback, and respond to their comments below.
>
> **Weak point 1:** The reviewers primary concern was the lack of explicit detail around proving that our approach satisfies the requirements of LDP. This was an oversight on our part, and we have now added multiple appendices with proofs as well as commentary within the body of the paper to further clarify.
>
> In particular, we have added an explicit definition for the local Laplace mechanism (see Equation 5), as well as the accompanying proof that the local Laplace mechanism satisfies epsilon-LDP in Appendix A (the proof that LDP is immune to post-processing is additionally provided in Appendix B). We have also updated Section 3.1 to clarify that sampling from $q_\phi(z|x)$ is equivalent to a Laplace mechanism $\mathcal{M}^\text{(local)}\left(x, \mu_{\phi}(\cdot), \epsilon_x \right)$, since sampling from $q_\phi(z|x)$ involves passing a datapoint through the mean function $\mu_\phi(x)$ and adding laplace noise of scale $b = \Delta\mu_\phi / \epsilon_x$, where $\Delta\mu_\phi$ is the sensitivity of $\mu_\phi(.)$, as determined by the clipping function.
>
> We now have all formal results in the paper needed to fully address the reviewers main concern, and hope that consequently the reviewer will consider raising their score.
>
> **Weak point 2:** For clarification, clean accuracy refers to the accuracy of our classifier when applied to a clean datapoint at inference time, while private accuracy refers to the accuracy of our classifier when applied to an LDP datapoint. In every case training is done on LDP data. In order to clarify these definitions to readers, the second paragraph of Section 4 has been updated significantly.
>
> With the proofs of privacy added and clarifications of the mechanism made in the text, we feel that we have addressed the primary concern of the reviewer. We hope they agree and will consider revising their score accordingly.

---

### Author Response · Authors · 2020-11-14
**Summary note**

We would like to thank each of the reviewers for their feedback.

Two common themes arose which we have addressed extensively. Firstly, it was requested that we add more rigorous proofs of how our approach produces $\epsilon$-LDP representations of data at both latent and feature level. These proofs have been included in Appendix A and B. Secondly, we have included a more thorough literature review, which outlines some recent existing work as highlighted by reviewers, and importantly, how our approach improves upon this work.

We feel that this review process has improved our paper and hope both the reviewers and readers of the paper agree.

---

### Decision · Program_Chairs · 2021-01-07
**Final Decision**

**Decision:**

Reject

**Comment:**

The paper considers the problem of private data sharing under local differential privacy.

(1) it assumes having access to a public unlabeled dataset for learning a VAE, so it reduces the dimensionality in a more meaningful way than simply running PCA. (2) the LDP guarantee is coming from the standard Laplace mechanism and Randomized Responses. (3) then the authors propose how to learn a model based on the privately released (encoded) data which exploits the knowledge of the noise distribution.

None of these components are new as far as I know, nor were they new in the context of differential privacy. For example, the use of a publicly available data for DP was considered in:

- Amos Beimel, Kobbi Nissim, and Uri Stemmer. Private learning and sanitization: Pure
vs. approximate differential privacy. In Approximation, Randomization, and Combinatorial Optimization. Algorithms and Techniques, pages 363–378. Springer, 2013.

(they called it Semi-Private Learning...)

- Papernot, N., Abadi, M., Erlingsson, U., Goodfellow, I., & Talwar, K. (2017). Semi-supervised knowledge transfer for deep learning from private training data. In ICLR-17.

The idea of integrating out the noise by leveraging the known noise structure were considered in:

- Williams, O., & McSherry, F. (2010). Probabilistic inference and differential privacy. Advances in Neural Information Processing Systems, 23, 2451-2459.

- Balle, B., & Wang, Y. X. (2018). Improving the Gaussian Mechanism for Differential Privacy: Analytical Calibration and Optimal Denoising. In International Conference on Machine Learning (pp. 394-403).

And many subsequent work.

The contribution of this work is in combining these known pieces (without citing some of the earlier work) to achieve a reasonably strong set of experimental results (for LDP standard).  I believe this is the first experimental study that uses VAE for the dimension reduction, however, this alone is not sufficient to carry the paper in my opinion; especially since the setting is now much easier, with access to a public dataset.

The reviewers question the experiments are baselines are usually not using a public dataset as well as the practicality of the proposed method.   Also, connections to some of the existing work on private data release (a.k.a., private synthetic data generation) were note clarified. For these reasons, there were not sufficient support among the reviewers to push the paper through.

The authors are encouraged to revise the paper according to the suggestions and resubmit in the next appropriate venue.